# Molecular Hybrids of Thiazolidinone: Bridging Redox Modulation and Cancer Therapy

**DOI:** 10.3390/ijms26136529

**Published:** 2025-07-07

**Authors:** Nourah A. Al Zahrani, Manal A. Alshabibi, Abrar A. Bakr, Fahad A. Almughem, Abdullah A. Alshehri, Huda A. Al-Ghamdi, Essam A. Tawfik, Laila A. Damiati

**Affiliations:** 1Department of Chemistry, College of Science, University of Jeddah, Jeddah 23218, Saudi Arabia; nalzahrani2@uj.edu.sa (N.A.A.Z.); halgamdi4@uj.edu.sa (H.A.A.-G.); 2Advanced Diagnostics and Therapeutics Institute, Health Sector, King Abdulaziz City for Science and Technology (KACST), Riyadh 11442, Saudi Arabia; malshabibi@kacst.gov.sa (M.A.A.); aabakr@kacst.gov.sa (A.A.B.); falmughem@kacst.gov.sa (F.A.A.); abdualshehri@kacst.gov.sa (A.A.A.); 3Department of Biological Science, College of Science, University of Jeddah, Jeddah 23218, Saudi Arabia

**Keywords:** thiazolidinone derivatives, biological activity, antioxidant properties, chemical synthesis, electron-donating groups

## Abstract

Heterocyclic compounds have shown that they hold significant therapeutic activities, highlighting the importance of discovering and developing novel candidates against cancers, infections, and oxidative stress-associated disorders. In this study, we demonstrated the biological activity of our previously synthesized thiazolidinone derivatives (TZDs-1, 6, and 7). Furthermore, we synthesized and structurally characterized a new derivative (TZD-5) using IR, ^1^H NMR, and ^13^C NMR spectroscopy, confirming the presence of its key functional groups, namely, carbonyl and imine. Their antioxidant activity was assessed through the DPPH assay, with TZD-5 showing the most potent effect (IC_50_ = 24.4 µg/mL), comparable to ascorbic acid, an effect attributed to the methoxy group introduced via N-alkylation. Cytotoxicity was evaluated using the MTS assay on normal (HFF-1) and cancerous (HepG2 and A549) cell lines at two time points: 24- and 48 h exposure. Our findings highlight clear differences in cytotoxicity and selectivity among the tested thiazolidinone derivatives. TZD-1 and TZD-6 demonstrated significant, dose-dependent cytotoxic effects on both cancerous (HepG2 and A549) and normal (HFF-1) cell lines, thus limiting their therapeutic potential due to insufficient selectivity. TZD-5 exhibited moderate selectivity with higher susceptibility for HepG2 cells compared to normal cells. Notably, TZD-7 showed the most favorable cytotoxic profile, demonstrating strong selective cytotoxicity toward cancer cell lines with minimal adverse effects on normal fibroblasts. Overall, the results highlight TZD-5 and TZD-7 as promising candidates for antioxidant and selective anticancer therapies.

## 1. Introduction

Antioxidants play a crucial role in mitigating oxidative stress, which results from an imbalance between reactive oxygen species (ROS) production and the body’s ability to neutralize these harmful molecules. Oxidative stress has been implicated in the pathogenesis of numerous conditions, including cardiovascular diseases, neurodegenerative disorders, and various types of cancer [1,2]. Interestingly, more than 75% of pharmaceuticals and natural herbal formulations incorporate heterocyclic moieties, many of which contain nitrogen and sulfur atoms [3]. These heterocyclic compounds have demonstrated significant therapeutic potential, including anti-tubercular [4], antimicrobial [5], anti-inflammatory [6], anti-diabetic [7], anticancer [8], and antiviral activities [9]. Such findings underscore the importance of exploring and developing heterocyclic compounds as promising candidates for combating cancers, emerging infectious diseases, and other oxidative stress-associated disorders.

Thiazolidinone is an azole ring with a sulfur atom at position 1; a nitrogen atom at position 3; and a carbonyl group at one of positions 2, 4, or 5. This unique structural framework has generated significant attention in medicinal chemistry due to its wide spectrum of biological activities, including antibacterial and cancer properties [10]. The pharmacological properties of thiazolidinone are highly dependent on its molecular structure, especially the nature of substituents on the core heterocyclic ring. The modifications applied to the structure, for instance, at positions 2, 4, and 5 may significantly influence the compound’s bioactivity [11]. Furthermore, the electron-withdrawing (such as nitro) or electron-donating (such as methyl) group may also enhance its antibacterial effectiveness [12].

Figure 1 illustrates several thiazolidinone derivatives (A–E) that demonstrate a wide array of biological activities [11,13]. Compounds with a 2-thiazolidinone core (A) have been studied as inhibitors of the BRD4 bromodomain [14], while derivatives with a 5-thiazolidinone scaffold (C) have applications in dye chemistry [15]. Additionally, 2-thioxo-4-thiazolidinones, also known as rhodanines, are essential structural components in numerous drug-like molecules. For instance, rosiglitazone, used in the treatment of type 2 diabetes, contains thiazolidinone (2,4-dione) pharmacophores. Overall, 4-thiazolidinones have been recognized as privileged structures with a wide range of pharmacological activities, including antitubercular, anti-inflammatory, antiviral, antimicrobial, antidiabetic, and antioxidant properties [16].

Despite their pharmacological versatility (Figure 2), the structure–activity relationships (SARs) of thiazolidinone derivatives are still not fully understood. To address this gap, we synthesized and characterized a series of thiazolidinone derivatives featuring different electronic properties: an electron-donating 4-methoxy group (compound **5**), an unsubstituted derivative (compound **6**), and an electron-withdrawing 4-nitro group (compound **7**). These minimal but informative variations allow us to explore preliminary SAR trends across multiple biological activities.

Building on our previously published work, where eight novel hybrid thiazolidinone derivatives containing gallic acid were synthesized through cyclocondensation and Knoevenagel reactions, we found that compound **5c** displayed superior antioxidant potential compared to ascorbic acid [17]. However, this current study seeks to highlight the therapeutic potential of thiazolidinone derivatives as promising candidates for applications as antioxidants and anticancer agents. This study will also focus on the SARs of novel thiazolidinone derivatives bearing electron-donating and electron-withdrawing substituents by evaluating their antioxidant potential and cytotoxic effects on both cancerous and non-cancerous cell lines. The findings aim to support the development of thiazolidinone-based therapeutics for combating cancers and oxidative stress-related diseases.

## 2. Results and Discussion

### 2.1. Chemical Profile of Thiazolidinone Derivative

Previously, thiazolidinone **1** was synthesized in good yield via a one-pot cyclocondensation reaction between 2-(3,4,5-trimethoxybenzylidene)hydrazine-1-carbothioamide and ethyl bromoacetate in the presence of sodium acetate. The **^1^H NMR** spectrum of compound 1 exhibited a characteristic singlet at δ = 3.87 ppm, corresponding to the CH_2_ group of the thiazolidinone ring. The **IR** spectrum displayed a prominent absorption band for the carbonyl group (C=O) at ν = 1704 cm^−1^. Additionally, the **^13^C NMR** spectrum of compound 1 showed ten carbon signals [17].

As illustrated in Figure 1, the target compound, 4-methoxy-substituted thiazolidinone 5, was synthesized in our previous study [17] through the reaction of thiazolidinone 1 with 4-methoxybenzyl bromide 2 in acetone in the presence of potassium carbonate under stirring for 24 h. The structure of compound 5 was confirmed by **IR**, **^1^H NMR**, and **^13^C NMR** spectroscopy. The **IR** spectrum of compound 5 displayed characteristic absorption bands at ν = 1727 cm^−1^ (C=O) and ν = 1634 cm^−1^ (C=N). The **^1^H NMR** spectrum revealed eight singlets at δ = 2.88, 3.85, 3.90, 3.93, 3.97, 4.39, 7.00, and 8.53 ppm, corresponding to two CH_2_ groups, four OCH_3_ groups, one aromatic proton, and one CH=N proton, along with two doublets for the aromatic ring protons. The **^13^C NMR** spectrum displayed sixteen carbon signals at δ = 33.0 (CH_2_), 56.0 (CH_2_), 56.1, 58.0, and 60.8 (OCH_3_), 105.1, 106.5, 128.7, 129.1, 131.1, 140.8, 153.4, 154.4, 157.8, and 172.3 ppm (C=O).

Thiazolidinone derivatives **6** and **7** were also synthesized as previously reported [17]. Their structures were confirmed by comparing the recorded spectral data with those in the literature. The **^1^H NMR** spectra of compounds **6** and **7** exhibited six singlets near δ = 3.8 ppm (CH_2_), 3.9 ppm (three OCH_3_), 5.0 ppm (CH_2_), 7.0 ppm (aromatic CH), and 8.3 ppm (CH=N), in addition to other expected aromatic proton signals. The **^13^C NMR** spectrum of compound **6** showed fifteen carbon signals, while that of compound **7** displayed fourteen signals, both consistent with previously reported data (Table 1). Furthermore, the **IR** spectra of compounds **6** and **7** were in full agreement with the published spectra [17].

### 2.2. Antioxidant Finding

In this study, the antioxidant activity of the newly synthesized thiazolidinone derivative **5** was evaluated, and its IC_50_ value was compared with those of the parent compound, thiazolidinone **1**; previously reported derivatives **6** and **7**; and the reference antioxidant, ascorbic acid. To ensure a valid comparison, all compounds were assessed under identical conditions using the DPPH (2,2-diphenyl-1-picrylhydrazyl) radical-scavenging assay.

Interestingly, our previous results indicated that thiazolidinone **1** and derivatives **6** and **7** exhibited relatively weak antioxidant activities, with IC_50_ values exceeding 120 µg/mL. In contrast, the new derivative, **5**, which incorporates an electron-donating methoxy group, demonstrated significantly enhanced antioxidant activity with an IC_50_ of 24.4 µg/mL, approaching that of ascorbic acid (IC_50_ = 11.76 µg/mL). These results suggest that the antioxidant activity of thiazolidinones can be markedly improved through the *N*-alkylation of lactam nitrogen in the thiazolidinone ring, particularly when incorporating electron-donating methoxy groups into the molecular framework (Figure 3, Figure 4 and Figure 5) [17].

The SAR method examines the relationship between a compound’s chemical structure and its biological activity. The objective of SAR is to identify the chemical group that is responsible for an organism’s biological response. Hence, these results can be explained either by single-electron transfer (SET) or by hydrogen atom transfer (HAT). In compounds with a high electron density, SET is expected, while HAT is expected in compounds with OH or NH groups that easily dissociate. Compared to compounds **1, 6**, and **7**, compound **5** has higher antioxidant activity that is structurally dependent. Furthermore, compound **5** exhibited antioxidant properties similar to those of ascorbic acid. According to the chemical structure of compound **5**, there are four powerful electron-donating groups (methoxy groups) that donate electrons from both sides to the central hydrazinothiazolidinone; these electrons stabilize free radicals. Moreover, this electron donation is crucial for the antioxidant mechanism, where the antioxidant molecule donates electrons to neutralize the free radical. This suggests that this compound has the highest electron density compared to others and would follow the SET mechanism.

A previous study by Lončarić et al. [18] reported that thiazolidinone derivatives bearing hydroxyl substituents on the phenyl ring exhibited strong inhibition of ABTS (2,2′-azinobis-(3-ethylbenzothiazoline-6-sulfonic acid)) radicals, implying that electron-donating groups such as hydroxyls enhance antioxidant activity. In contrast, methoxy-substituted derivatives in their study showed reduced activity. Another study by Liu et al. illustrated that the novel 5-benzylidenethiazolidine-2,4-dione and 5-(furan-2-ylmethylene)thiazolidine-2,4-dione play an important role as inhibitors of insulin-like growth factor-1 receptor (IGF-1R) through hierarchical virtual screening. SAR and biological evaluation showed that these compounds had excellent potency (with IC_50_ values of 57 and 61 nM, respectively) and selectivity over the insulin receptor [19]. These observations are consistent with our findings, highlighting that while electron-donating groups generally enhance antioxidant effects, their position and interaction with the molecular scaffold are crucial; in some cases, electron-withdrawing groups may diminish antioxidant efficacy.

### 2.3. Cell Viability Assessment

The MTS assay was employed in the current study to evaluate the in vitro effects of thiazolidinone derivatives (TZDs-1, 5, 6, and 7) on the cellular viability of the HFF-1, HepG2, and A549 cell lines following incubation for 24 and 48 h. Using the MTS assay allows for the quantitative measurement of how TZDs affect cell viability after a certain period of treatment. The 24- and 48-hour incubation periods are useful for observing either immediate and delayed cytotoxicity or effects on cell growth, providing more comprehensive data on the potential therapeutic or toxic properties of thiazolidinone derivatives.

The cytotoxic effects of the thiazolidinone derivative (TZD-1) were assessed at different concentrations over 24- and 48-hour periods, and the results are presented in Figure 6. The primary objective was to determine whether TZD-1 exhibits selective cytotoxicity toward cancer cells while sparing normal cells. From the presented result, the lack of a viability difference between normal and cancerous cell lines at cytotoxic concentrations indicates that TZD-1 does not exhibit selectivity toward cancer cells and may pose toxicity risks to normal tissues. For instance, analysis of cell viability data revealed that HepG2 and A549 cell viability dropped below 80% at a concentration of ≥1250 µg/mL, indicating a significant cytotoxic effect on cancer cells. Specifically, at 24 h (Figure 6), HepG2 cellular viability was reduced to 53.16%, and A549 cellular viability was 75.58%, while at 48 h (Figure 6), the cellular viability of these cancer cell lines further decreased to 21.91% and 70.64%, respectively. However, HFF-1 cell viability at the same concentration was substantially reduced to 6.45% at 24 h and 15.34% at 48 h, suggesting that normal cells were equally or even more susceptible to the cytotoxic effects of TZD-1. Overall, a dose-dependent decrease in viability was observed in cancer cell lines, while normal cells (HFF-1) showed higher resistance, indicating the potential selectivity of TZD-1 toward cancer cells (Figure 6). However, its potential for clinical application may be limited unless modifications can be implemented to improve its selectivity.

The cytotoxicity of the second thiazolidinone derivative (TZD-5) was evaluated against HFF-1 and the two cancerous cell lines over 24 and 48 h as well (Figure 7). An analysis of the cell viability data showed that HepG2 dropped to below 80% at a concentration of ≥2500 µg/mL (HepG2 = 63%) after 24 h (Figure 7), indicating its cytotoxic effects at high doses. However, at these same concentrations, HFF-1 cell viability was 97% at 2500 µg/mL, suggesting that normal cells were relatively more resistant to TZD-5 than the hepatic cancer cells used. This trend became even more pronounced at 48 h (Figure 7), where HepG2 cellular viability dropped to 57% at 2500 µg/mL, while HFF-1 retained 48% viability at the same concentration. These findings suggest that TZD-5 exhibits greater cytotoxicity toward hepatic cancer cells than normal cells, particularly at higher concentrations and longer exposure times, indicating some degree of selectivity. However, the reduction in HFF-1 viability at very high concentrations suggests potential off-target effects, which could limit its therapeutic application. Generally, the results showed a concentration- and time-dependent cytotoxic response, with HepG2 cells being relatively more sensitive compared to normal HFF-1 and A549 cancer cells, especially at higher concentrations (Figure 7). Further studies should focus on optimizing the selectivity of TZD-5 through structural modifications or targeted delivery approaches to enhance its anticancer potential while minimizing toxicity to normal cells.

The cytotoxic effects of the thiazolidinone derivative (TZD-6) were evaluated to assess its potential selectivity, similar to the previous compounds, and the result is presented in Figure 8. At 24 h, HepG2 and A549 cellular viability reduced below 80% at 156 µg/mL (HepG2 = 76%, A549 = 78%), indicating the compound’s effectiveness in reducing cancer cell viability as presented in Figure 8. As the concentration increased, both HepG2 and A549 showed a further drop in viability, reaching 25% and 13% at 1250 µg/mL and 35% and 10% at 5000 µg/mL, respectively. However, HFF-1 viability also reduced significantly, with a sharp decrease observed at 312 µg/mL (70%) and further decreasing to 2% at 1250 µg/mL, indicating that the normal cells were highly sensitive to TZD-6. At 48 h (Figure 8), the cytotoxic effects became even more obvious, with HepG2 and A549 viability decreasing to 12% and 9% at 1250 µg/mL and 7% and 13% at 2500 µg/mL, while HFF-1 viability was nearly depleted at these concentrations (1–2%). TZD-6 exhibits strong concentration- and time-dependent cytotoxicity against all tested cell lines, with significant reductions in viability observed at higher concentrations, showing limited selectivity between cancerous and normal cells (Figure 8). The substantial toxicity to normal fibroblasts suggests that further optimization is required to improve its selectivity, possibly through structural modifications or targeted delivery systems to minimize adverse effects on non-cancerous cells while maintaining its anticancer efficacy.

The effects of the TZD-7 compound on cell viability were also evaluated, and the data are exhibited in Figure 9. At 24 h (Figure 9), HFF-1 cells showed high viability across concentrations ≤1250 µg/mL, ranging from 135% at 39 µg/mL to 110% at 1250 µg/mL, demonstrating resistance to the compound even at higher concentrations. In contrast, A549 cancer cells displayed reduced viability at higher concentrations, with cellular viability of 70% at the same concentration. After 48 h (Figure 9 (right)), the presented data showed a significant reduction in the cell viability of hepatic cancer cells (HepG2) to around 23% following the application of 1250 µg/mL, while the normal cells exhibited cellular viability of 94%. Overall, TZD-7 demonstrates selective cytotoxicity against cancer cells, with increasing potency over time, while sparing non-cancerous cells, particularly at lower concentrations (Figure 9).

Overall, the cytotoxicity assessment of the thiazolidinone derivatives revealed distinct differences in selectivity and potency between the tested compounds. TZD-1 and TZD-6 exhibited significant cytotoxic effects on HepG2 and A549 cancer cells, with viability dropping below 25% at concentrations of ≥1250 µg/mL; however, the normal HFF-1 cells were equally or more susceptible, with viability decreasing to as low as 6.45% at 24 h, limiting their therapeutic potential. TZD-5 showed moderate selectivity, as HepG2 cell viability dropped to 57% at 2500 µg/mL while the HFF-1 cells retained 48% viability, indicating a degree of preferential toxicity toward cancer cells, though off-target effects at high doses remain a concern. In contrast, TZD-7 exhibited the highest selectivity, with HFF-1 cell viability remaining above 94% even at 1250 µg/mL, while HepG2 cell viability dropped significantly to 23% at 48 h. These findings suggest that while all compounds demonstrated anticancer potential, TZD-7 holds the most promise for further development (Table 2), with future studies needed to optimize its selectivity and minimize toxicity through structural modifications or targeted delivery strategies.

These findings are consistent with Deng et al., whose study showed a thiazolidinone compound (**12i**) exhibited significant activity against the HepG2 cell line, with an IC_50_ value of 4.40 μM and A549 with an IC_50_ value of 10.06 μM [20]. Furthermore, Sethi et al. showed that TZ-5 and TZ-13 exhibited good antiproliferative activity against the A549 cancer cell line, while TZ-10 exhibited moderate antiproliferative activity against the HepG2 cell line [21]. Another study by Hanafy et al. developed and evaluated novel thiazolidine-2,4-diones as dual inhibitors of EGFR (epidermal growth factor receptor) and VEGFR-2 (vascular endothelial growth factor) against various cancer cell lines, including HCT-116, MCF-7, A549, and HepG2. They found that compounds 5g and 4g exhibited strong anticancer activities against the HepG2 and A549 cell lines, with IC_50_ values of 3.86 μM and 7.55 μM, respectively. These compounds also showed favorable ADMET (absorption, distribution, metabolism, excretion, and toxicity) profiles [22]. In addition, hybrid molecules combining thiazolidinone and thiosemicarbazone (TZD-TSC) moieties were synthesized and tested for anticancer activity. TZD-TSC 3 demonstrated high toxicity toward HepG2 cells with IC50 values of 2.97 ± 0.39 µM and IC50 values of 28.34 ± 2.21 µM against human glioblastoma T98G cells, indicating a strong preference for targeting cancer cells over normal cells [23]. These reported studies support our findings on the selective cytotoxicity of certain TZD derivatives.

## 3. Materials and Methods

### 3.1. Materials

All solvents used in the preparation of thiazolidinone derivatives were obtained in pure form from Sigma-Aldrich (St. Louis, MO, USA) and Thermo Fisher Scientific (Waltham, MA, USA). All other chemicals used herein were of analytical grade and were used without further purification. To separate and confirm the purity of thiazolidinone derivatives, column chromatography on silica gel (0.063–0.2 mm) and thin layer chromatography (TLC) with aluminum silica gel F254 were used. Human foreskin fibroblasts (HFF-1) (ATCC: SCRC-1041), hepatoblastoma cell line (HepG2) (ATCC: HB-8065), and human lung cancer cell line (A549) (ATCC: CCL-185) were used in this study to evaluate the cytotoxicity of the tested compounds. Distilled water for experiment preparation was generated using a Milli-Q^®^ IQ 7005 Purification System (Millipore SAS, Molsheim, France).

### 3.2. Chemistry

#### 3.2.1. Synthesis of 3-(4-methoxybenzyl)-2-((3,4,5-trimethoxybenzylidene)hydrazineylidene) thiazolidin-4-one (**5**)

In a round flask, Compound 1 (2-(2-(3,4,5-trimethoxybenzylidene) hydrazineyl)thiazol-4(5H)-one) (0.31 gm, 1 mmol), which prepared previously in our work [17] was dissolved/suspended in 20 mL of anhydrous acetone in the presence of anhydrous potassium carbonate (0.39 gm, 2.8 mmol); then, 4-methoxy benzyl bromide (0.24 gm, 1.4 mmol) was added under stirring for 24 h (TLC monitoring, with petroleum ether/ethyl acetate, 6:4, *v*/*v*). The mixture was poured on ice and filtered to obtain compound **5** in high yield as a pale-yellow powder.

#### 3.2.2. Nuclear Magnetic Resonance (NMR) Analysis

**^1^H** and **^13^C NMR** spectra were recorded in CDC_l3_ on a Bruker Avance 500 MHz spectrometer. The reported chemical shifts were against TMS. **^1^H NMR** (500 MHz, CDCl_3_) δ (ppm): 2.88 (s, CH_2_, 2H), 3.85 (s, OCH_3_, 3H), 3.90 (s, OCH_3_, 3H), 3.93 (s, OCH_3_, 3H), 3.97 (s, OCH_3_, 3H), 4.39 (s, CH_2_, 2H), 7.0 (s, Ar-H, 2H), 7.18 (d, J = 7 Hz, Ar-H, 2H), 7.45 (d, J = 7 Hz, Ar-H, 2H), and 8.53 (s, H–C=, 1H). **^13^C NMR** δ (ppm): 33.0, 56.0, 56.1, 58.0, 60.8, 105.1, 106.5, 128.7, 129.1, 131.1, 140.8, 153.3, 153.4, 154.4, 157.8, and 172.3 (Appendix A).

#### 3.2.3. Fourier-Transform Infrared Spectroscopy (FTIR) Analysis

FTIR spectra were recorded using a Thermo Scientific Nicolet iS10 FTIR spectrometer (Thermo Fisher Scientific, Waltham, MA, USA). The synthesized thiazolidinone derivatives were placed as KBr discs and were analyzed at a wavenumber range of 4000 to 500 cm^−1^. Melting points were determined in open capillary tubes in the Stuart Scientific melting point apparatus SMP3 and were uncorrected. **IR** cm^−1^: 3031 (C–H sp^2^), 2933(C–H sp^3^), 1727 (C=O), 1634 (C=N), 1570, 1506 (C=C), 1234 (C–N), 1125 (C–O), and 832 (C–H olefinic) (Appendix A).

#### 3.2.4. Antioxidant Activity

The antioxidant activity of thiazolidinone derivative **5** was evaluated under conditions similar to those reported in our previous work for thiazolidinone derivatives (**1**, **6**, and **7**) using the DPPH assay (2,2-diphenyl-1-picrylhydrazyl) [17]. The concentration needed to get rid of 50% of free radicals is called IC_50_ (half-maximal inhibitory concentration) (μg/mL). The value of the IC_50_ is determined by plotting antioxidant activity against the concentration employing the equation below:%Antioxidant activity = (A_control_ − A_sample_/A_control_) × 100
where A_control_ is the absorbance of the DPPH solution without the sample, and A_sample_ is the absorbance with the tested compound. These percentage inhibition values are then plotted against the logarithm of the compound concentrations to generate a dose–response curve.

The IC_50_ value, which represents the concentration required to inhibit 50% of the DPPH radicals, is determined from this curve using nonlinear regression analysis (or linear interpolation when appropriate).

### 3.3. Cell Viability Assay

HFF-1, HepG2, and A549 were cultured in DMEM (Sigma-Aldrich, St. Louis, MO, USA) medium complemented with 10% fetal bovine serum (FBS) (Sigma, Suffolk, UK), streptomycin (100 mg/mL), and penicillin (100 units/mL) (Sigma, Suffolk, UK) in a cell culture incubator (5% CO_2_ at 37 °C).

Thiazolidinone derivatives (TZDs-**1**, **5**, **6**, and **7**) were prepared via dissolution in 10% DMSO and 8 serial dilutions and assessed starting from 5000 to 39 µg/mL. A CellTiter 96^®^ AQueous One Solution Cell Proliferation Assay (MTS kit, Promega, Madison, WI, USA) was used to evaluate the cell viability of TZDs on the cells. Briefly, all cells were seeded into 96-well plates with a seeding density of 1.5 × 10^4^ cells per well. After 24 h, cells adhered, and then, TZD derivatives were added, and plates were incubated in a cell culture incubator with 5% CO_2_ at 37 °C for 24 and 48 h. Completed DMEM was used as the positive control, and 0.1% Triton X-100 was used as the negative control. After incubation time, cells were washed with sterile phosphate-buffered saline, pH 7.4 (PBS), and then, a mixture of complete DMEM medium and MTS reagent was added to each well (100 µL of DMEM medium + 20 µL of MTS reagent). Finally, the plates were incubated for 3 h in a cell culture incubator. The absorbance at 492 nm was recorded using a Cytation ™ 3 Cell Imaging Multi-Mode Reader (BIOTEK Instruments Inc., Winooski, VT, USA). The percentage of cellular viability was calculated using the following equation:Cell Viability (%) = (S − T)/(H − T) × 100
where S is the absorbance of the cells treated with the applied compounds, T is the absorbance of the positive control, and H is the absorbance of the negative control. The results are presented as the mean ± SD of at least three independent measurements.

### 3.4. Statistical Analysis

Experiments were performed as three independent replicates, and the results were expressed as the mean ± standard deviation (SD). Statistical analysis was performed using GraphPad Prism v.10 (GraphPad Software, La Jolla, CA, USA). Statistical significance was set at *p* ≤ 0.05.

## 4. Study Limitations

This study has a few limitations. Although compounds TZD-1 and TZD-6 exhibited potent anticancer activities, their lack of selectivity toward cancer cells, indicated by substantial toxicity to normal cells, limits their therapeutic application. Additionally, more cell lines need to be tested to achieve broader screening for these compounds. Lastly, although SAR was discussed, a more extensive SAR exploration involving varied substituents could further clarify optimal structural requirements for enhanced biological activities. Addressing these limitations through future studies will provide deeper insights and strengthen the therapeutic potential of the synthesized thiazolidinone derivatives.

## 5. Conclusions

This study successfully synthesized and structurally characterized different thiazolidinone derivatives (compounds **1**, **5**, **6**, and **7**) using **IR** spectroscopy, **^1^H NMR**, and **^13^C NMR** techniques. Among the synthesized compounds, derivative 5, featuring a methoxy-substituted phenyl ring, exhibited markedly improved antioxidant activity, with an IC_50_ value of 24.4 µg/mL. This activity was significantly higher than that of its structural analogs and precursors (IC_50_ > 120 µg/mL), underscoring the beneficial influence of electron-donating substituents on radical scavenging capacity. In vitro cytotoxicity assays against human hepatocellular carcinoma (HepG2) and lung adenocarcinoma (A549) cell lines indicated that TZD-1 and TZD-6 possess potent anticancer activity; however, their high toxicity toward normal human fibroblast cells (HFF-1) poses a limitation for therapeutic application due to a lack of selectivity. Notably, compound TZD-7 demonstrated promising selective cytotoxicity by significantly inhibiting the proliferation of cancer cells while maintaining minimal toxicity toward normal fibroblasts. These findings highlight TZD-7 as a strong candidate for further preclinical evaluation as a selective anticancer agent. Overall, the results contribute valuable insights into the SARs of thiazolidinone-based scaffolds, particularly regarding antioxidant and anticancer activities. Future studies focused on structure refinement and mechanism elucidation are warranted to fully realize the therapeutic potential of these compounds, with TZD-7 emerging as a lead candidate for targeted anticancer drug development.

## Data Availability

The original contributions presented in the study are included in the article; further inquiries can be directed to the corresponding authors.

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
