# Peer review of "Molecular Hybrids of Thiazolidinone: Bridging Redox Modulation and Cancer Therapy"

_ijms, 2025, doi:10.3390/ijms26136529_

Round 1
Reviewer 1 Report (New Reviewer)
Comments and Suggestions for Authors
The manuscript titled "Molecular Hybrids of Thiazolidinone: Bridging Redox Modulation and Cancer Therapy" (ijms-3748658) is well written and meets the standards of the journal. It presents interesting and valuable findings for the entire group of compounds studied, with particularly promising results for TZD-7.
However, there are some typographical errors throughout the text that should be carefully reviewed and corrected. Additionally, a graphical abstract clearly illustrating the scope and outcome of the study would be a welcome addition and would enhance the overall presentation of the work.
Author Response
Attached

Reviewer 2 Report (New Reviewer)
Comments and Suggestions for Authors
This study investigated a new thiazolidinone derivative (TZD-5), along with three previously synthesized compounds (TZDs-1, 6, and 7). The research involved structural characterization, analysis of antioxidant activity, and evaluation of cytotoxic effects against normal (HFF-1) and cancerous (HepG2, A549) cell lines. The results identified TZD-5 and TZD-7 as promising candidates for antioxidant and selective anticancer therapies. However, several major and minor issues within the manuscript require revision to enhance clarity, accuracy, and consistency.
Major issues:
- Please check the usage of 'thiazolidinone' and 'thiazolidinedione' throughout the full text to ensure consistency.
- In the Introduction section, the sentence “Furthermore, the electron-donating (such as nitro) or electron-withdrawing (such as methyl) group, may also enhance the antibacterial effectiveness [12],” contains a scientific error. The nitro group (-NOâ‚‚) is a classic electron-withdrawing group, not an electron donor. It reduces the electron density of a benzene ring or other structures through its inductive effect (-I) and mesomeric effect (-M). Conversely, the methyl group (-CH₃) is an electron-donating group, increasing electron density via hyperconjugation (+σ) and a weak inductive effect (+I).
- In section ‘2.2.4 Antioxidant Activity’ , while the method for evaluating antioxidant activity using ICâ‚…â‚€ is introduced, the explanation of the relationship between the ICâ‚…â‚€ value and antioxidant activity remains unclear. Furthermore, the provided equation only addresses the calculation of antioxidant activity, without clearly correlating it to the determination of ICâ‚…â‚€. A more explicit explanation of how ICâ‚…â‚€ is derived from the antioxidant activity percentages is needed.
- In the third paragraph of section ‘3.2. Antioxidant Finding’, the conclusion that compound 5’s structure contributes to its antioxidant function is stated. However, the logical coherence of this paragraph needs improvement to clearly articulate how the described structural features (e.g., methoxy groups and electron density) lead to the observed antioxidant activity.
Minor issues:
- In the Abstract section, the phrase “In contrast” is not appropriate and should be changed to “Additionally” or “Furthermore.” This is because the synthesis of TZD-5 applies to current thiazolidinone derivatives, rather than contrasting with them.
- In the ‘2.1 Materials’ section, the sentence “Human foreskin fibroblasts (HFF-1) (ATCC: SCRC-1041), hepatoblastoma cell line (HepG2) (ATCC: HB-8065), and human lung cancer cell line (A549) (ATCC: CCL-185),” is incomplete as it lacks a subject and a predicate.
- In the ‘3 Cell Viability Assay’, "For 'HepG-2' should be consistent with 'HepG2' as used in previous sections (Abstract, Materials)."
- Page 4/19, line 4: “product 1” is mentioned without prior definition. Please clarify what this refers to, as it appears to be the first time this specific product or thiazolidinone is introduced.
- On page 5/19, line 7, in the equation for Cell Viability (%) , the calculation appears to be missing a multiplication symbol for "100" and should explicitly be "
Round 2
Reviewer 2 Report (New Reviewer)
Comments and Suggestions for Authors
I have reviewed the author's revisions and their response to my previous comments. The author has addressed all points raised.
The manuscript is now significantly improved, and I have no further concerns. I recommend it be accepted for publication in its current form.
This manuscript is a resubmission of an earlier submission. The following is a list of the peer review reports and author responses from that submission.
Round 1
Reviewer 1 Report
Comments and Suggestions for Authors
This study aims to synthesize, characterize, and perform biomedical evaluations of thiazolidinone derivatives (TZDs-1, 5, 6, and 7). Some critical concerns should be addressed as follows:
- The authors mentioned in the abstract that they characterized the derivatives using IR, ¹H NMR, and ¹³C NMR spectroscopy, but I did not find any data, including figures and charts for these findings.
- The antibacterial data showed negative results for all compounds; thus, I recommend removing these data and insinuate about these limitations in future perspectives.
- Abstract: it should be improved, highlighting the key findings and methods applied in this study.
- Methods: The authors used DPPH assay and I recommend to conduct ABTS for confirming these results. How did you calculate IC50? Antioxidant methods should be described in details. Line 150: is the density of cells 1.5 ×104 right? Statistical analysis: The authors should perform statistical analysis using a t-test or ANOVA, according to the raw data, and normalize the data before analysis.
- Results and discussion: The authors should elucidate the antioxidant attributes of the compounds in light of their chemical structures. The cytotoxicity results should be discussed and supported by previous literature, as they seem to be presented without discussion. Antibacterial findings should be removed. Limitations of this study should be discussed.
Author Response
We thank the reviewers for their thoughtful and helpful comments. We respond to each point below; their suggestions have undoubtedly improved the paper, for which we are very grateful. The new sections are highlighted in blue font in the revised manuscript.
Reviewer 1
This study aims to synthesize, characterize, and perform biomedical evaluations of thiazolidinone derivatives (TZDs-1, 5, 6, and 7). Some critical concerns should be addressed as follows:
Comment: The authors mentioned in the abstract that they characterized the derivatives using IR, ¹H NMR, and ¹³C NMR spectroscopy, but I did not find any data, including figures and charts for these findings.
Response: Sorry for the confusion. As highlighted in the manuscript, we have utilized some of the derivatives that were synthesized in our previous study but were not tested in vitro, and we synthesized a new derivative (TZD-5) which we chemically analyzed. We added the NMR and IR results for that derivative in the supplementary materials section - Figurers 1 - 3. Thank you for highlighting this.
Comment: The antibacterial data showed negative results for all compounds; thus, I recommend removing these data and insinuate about these limitations in future perspectives.
Response: Thank you for your comment. Since TZDs have shown in previous studies that they have potential antibacterial activities against different bacterial strains, while the studied derivatives did not, we wanted to highlight this so further investigations can focus on structural modification and improving their effectiveness against bacteria. However, to comply with the second part of this comment, we decided to remove the Figure, as it will not add anything, but kept the text to give the readers some authentication that we tested the compound using a Gram-positive and a Gram-negative bacterium up to the toxic concentration (5000 µg/mL) and there was no effect. We also discussed some previous studies that showed the antibacterial effects of their thiazolidinone derivatives. This also indicates that every derivative has a specific biological effect, such as the ones in this current study, which demonstrated anticancer activity but not antibacterial.
Comment: Abstract: it should be improved, highlighting the key findings and methods applied in this study.
Response: Thank you for your comment. We enhanced the clarity and structure of the abstract. We also restructured the introduction by moving some paragraphs to enhance the clarity and the text flow. Changes are in blue.
Comment: Methods: The authors used DPPH assay and I recommend to conduct ABTS for confirming these results. How did you calculate IC50? Antioxidant methods should be described in details. Line 150: is the density of cells 1.5 ×104 right? Statistical analysis: The authors should perform statistical analysis using a t-test or ANOVA, according to the raw data, and normalize the data before analysis.
Response: Thank you for your valuable comment. Regarding your inquiry about the use of the DPPH method for evaluating antioxidant activity, we would like to clarify that we specifically chose the DPPH assay because it is a well-established, reliable, and widely accepted method for assessing free radical scavenging activity. Furthermore, the primary objective of our current study was to in vitro evaluate the synthesized derivatives previously published along with synthesizing and evaluating one new derivative that could hold high potential in biological application in future. It showed that this newly synthesized derivative (TZD-5) had a moderate selectivity with higher susceptibility of HepG2 cells compared to the normal cell line (HFF-1).
Since the antioxidant activity of the original compounds had already been evaluated using the DPPH method in our earlier publication, we opted to use the same method for consistency and comparability. This allows for a more direct and meaningful comparison between the new derivative and the previously studied compounds under the same experimental conditions. For this reason, we did not include additional antioxidant assays, as our aim was not to comprehensively re-evaluate the antioxidant profile of the compound class but rather to assess the relative improvement or modification in activity due to structural derivatization.
Regarding the cell seeding, yes, we confirm that the seeding density was 1.5 ×104 / well, as we usually seed between 10,000 to 15,000 cells / well in the 96-well plates.
For the statistical analysis, we used One-Way ANOVA test, and new figures were added in the supplementary materials section - Figurers 4 to 7.
Comment: Results and discussion: The authors should elucidate the antioxidant attributes of the compounds in light of their chemical structures. The cytotoxicity results should be discussed and supported by previous literature, as they seem to be presented without discussion. Antibacterial findings should be removed. Limitations of this study should be discussed.
Response: Thank you for your comment. A new section has been added to address the antioxidant attributes of the SAR that can be found under section 3.2 – blue texts. The cytotoxicity section has been thoroughly discussed, with new references included, which can be found under section 3.3 – blue texts. In addition, the antibacterial section has been removed, as explained in the second comment. The study’s limitations have also been incorporated before the conclusion section under section 4.
Reviewer 2 Report
Comments and Suggestions for Authors
The manuscript authored by Alzahrani to focuses on the activity of thiazolidinedione based ibrid and their properties as antioxidants and anticancer agents. It also mentions their antibacterial properties, although I could not find a clear explanation for this aspect. Indeed, this is not even reflected in the title.
My primary concern is that the number of synthesized derivatives is rather limited. Although they are clearly characterized, the quality and scope of the research do not seem sufficient to meet the standards of the journal. Moreover, compound characterization should be considered a standard requirement and not presented as an exceptional achievement.
Based on the data provided, I believe it is not feasible to conduct a meaningful SAR study or to draw solid conclusions.
Author Response
We thank the reviewers for their thoughtful and helpful comments. We respond to each point below; their suggestions have undoubtedly improved the paper, for which we are very grateful. The new sections are highlighted in blue font in the revised manuscript.
Comment: The manuscript authored by Alzahrani to focuses on the activity of thiazolidinedione based ibrid and their properties as antioxidants and anticancer agents. It also mentions their antibacterial properties, although I could not find a clear explanation for this aspect. Indeed, this is not even reflected in the title.
Response: Thank you for your comment, this was also highlighted in the second reviewer. Since TZDs have shown in previous studies that they have potential antibacterial activities against different bacterial strains, while the studied derivatives did not, we wanted to highlight this so further investigations can focus on structural modification and improving their effectiveness against bacteria. However, to comply with this comment, we decided to remove the Figure, as it will not add anything, but kept the text to give the readers some authentication that we tested the compound using a Gram-positive and a Gram-negative bacterium up to the toxic concentration (5000 µg/mL) and there was no effect. We also discussed some previous studies that showed the antibacterial effects of their thiazolidinone derivatives. This also indicates that every derivative has a specific biological effect, such as the ones in this current study, which demonstrated anticancer activity but not antibacterial.
Comment: My primary concern is that the number of synthesized derivatives is rather limited. Although they are clearly characterized, the quality and scope of the research do not seem sufficient to meet the standards of the journal. Moreover, compound characterization should be considered a standard requirement and not presented as an exceptional achievement.
Response: The importance of exploring and developing heterocyclic compounds as promising candidates for combating cancers, emerging infectious diseases and other oxidative stress-associated disorders is on the rise. We have utilized some of the derivatives that were synthesized in our previous study, which are novel but were not tested in vitro and it is essential to demonstrate their biological activity. We also synthesized a new derivative (TZD-5) which we chemically analyzed, and we added the NMR and IR results as supplementary Figures 1-3, it exhibited a moderate selectivity with higher susceptibility of HepG2 cells compared to the normal cell line (HFF-1), which could hold high potential in biological application in future. Thank you for highlighting this.
Comment: Based on the data provided, I believe it is not feasible to conduct a meaningful SAR study or to draw solid conclusions.
Response: Thank you for your valuable comment. Although the number of synthesized derivatives tested is limited, our findings highlight distinct patterns in biological activities, particularly regarding antioxidant and anticancer properties. Derivative TZD-5 exhibited significantly enhanced antioxidant activity compared to other tested derivatives, indicating the importance of electron-donating substituents in radical scavenging capacity. Similarly, the differential cytotoxic effects observed, especially the selective cytotoxicity of derivative TZD-7, provide preliminary but meaningful insights into structure-activity relationships. A new section has been added to emphasize this part.
“The SAR method examines the relationship between a compound's chemical structure and its biological activity. The objective of SAR is to identify the chemical group that is responsible for an organism's biological response. Hence, these results can be explained either by single electron transfer (SET) or by hydrogen atom transfer (HAT). In compounds with a high electron density, SET is expected, while HAT is expected in compounds with OH or NH groups that easily dissociate. Compared to compounds 1, 6 and 7, compound 5 has a higher antioxidant activity that is structurally dependent. Furthermore, compound 5 exhibited antioxidant properties similar to those of ascorbic acid. According to the chemical structure of compound 5, there are four powerful electron donating groups (methoxy group) that donate electrons from both sides to the central hydrazinothiazolidinone, which suggests that this compound has the highest electron density compared to others and would follow SET”
However, we acknowledge that to strengthen and expand these SAR conclusions, further structural optimization and testing of a broader range of compounds with systematic modifications are necessary. Future studies are planned to build upon these initial findings and achieve more robust and comprehensive SAR analyses.
Overall, with the valuable comment of you and the second reviewer, this revised manuscript was improved by first enhancing the clarity and structure of the abstract, restructuring the introduction by moving some paragraphs to improve the clarity and the text flow, adding a thorough statistical analysis (using one-way ANOVA) for the cell viability assessment, and including a study limitation section (section 4). We also included the NMR and the IR Figures for the newly synthesized derivative (TZD-5) in the Supplementary Materials section. We hope that these changes have overcome some of your concerns.
Round 2
Reviewer 1 Report
Comments and Suggestions for Authors
The authors addressed all concerns to full satisfaction. From my point of view, the current form of this manuscript is suitable for publication.